# Structural Changes in Hippocampal Subfields in Patients with Continuous Remission of Drug-Naive Major Depressive Disorder

**DOI:** 10.3390/ijms21093032

**Published:** 2020-04-25

**Authors:** Asuka Katsuki, Keita Watanabe, LeHoa Nguyen, Yuka Otsuka, Ryohei Igata, Atsuko Ikenouchi, Shingo Kakeda, Yukunori Korogi, Reiji Yoshimura

**Affiliations:** 1Department of Psychiatry, University of Occupational and Environmental Health, 1-1 Iseigaoka, Yahatanishi-ku, Kitakyushu 807-8555, Japan; asuka-k@med.uoeh-h.ac.jp (A.K.); leehoa2k4@med.uoeh-u.ac.jp (L.N.); yharada@med.uoeh-u.ac.jp (Y.O.); igataryouhei@gmail.com (R.I.); atsuko-i@med.uoeh-u.ac.jp (A.I.); 2Department of Radiology, University of Occupational and Environmental Health, 1-1 Iseigaoka, Yahatanishi-ku, Kitakyushu 807-8555, Japan; sapient@med.uoeh-u.ac.jp (K.W.); ykorogi@med.uoeh-u.ac.jp (Y.K.); 3Department of Radiology, Hirosaki University Graduate School of Medicine, Hirosaki 036-8562, Japan; kakeda@med.uoeh-u.ac.jp

**Keywords:** brain morphology, major depressive disorder, hippocampus, cornu Ammonis, dentate gyrus, antidepressant

## Abstract

Objective: Hippocampal volume is reduced in patients with major depressive disorder (MDD) compared with healthy controls. The hippocampus is a limbic structure that has a critical role in MDD. The aim of the present study was to investigate the changes in the volume of the hippocampus and its subfields in MDD patients who responded to antidepressants and subsequently were in continuous remission. Subjects and Methods: Eighteen patients who met the following criteria were enrolled in the present study: the DSM-IV-TR criteria for MDD, drug-naïve at least 8 weeks or more, scores on the 17-items of Hamilton Rating Scale for Depression (HAMD) of 14 points or more, and antidepressant treatment response within 8 weeks and continuous remission for at least 6 months. All participants underwent T1-weighted structural MRI and were treated with antidepressants for more than 8 weeks. We compared the volumes of the hippocampus, including its subfields, in responders at baseline to the volumes at 6 months. The volumes of the whole hippocampus and the hippocampal subfields were measured using FreeSurfer v6.0. Results: The volumes of the left cornu Ammonis (CA) 3 (*p* = 0.016) and the granule cell layer of the dentate gyrus (GC-DG) region (*p* = 0.021) were significantly increased after 6 months of treatment compared with those at baseline. Conclusions: Increases in volume was observed in MDD patients who were in remission for at least 6 months.

## 1. Introduction

Major depressive disorder (MDD) is a lifelong, episodic, prevalent and disabling mental disorder found in individuals worldwide. The cause of MDD is multifactorial, including genetic, neurobiological, and environmental factors, as well as their interactions. The hippocampus plays an important role in MDD. According to the reviewed literature, MDD is characterized by an altered structural network that encompasses reduced volumes of the orbitofrontal cortex(OFC), anterior cingulate, hippocampus, and striatum [1]. The anterior cingulate cortex, amygdala and hippocampus comprise an interconnected prefrontal neocortical and limbic network that is dysregulated in MDD. Modulation of this prefrontal–limbic network occurs primarily through the hypothalamus, basal ganglia and midbrain [2]. Structural abnormalities in the hippocampus are present in MDD. This region is considered to regulate behavioral and neuroendocrine responses to stress and can be damaged by excessive exposure to the stress-induced release of steroidal and inflammatory signaling molecules [3]. The hippocampus is a complex structure and is related to many emotional, memory, and cognitive functions. MDD is associated with a volume reduction in the hippocampus compared with healthy controls, a finding that has been relatively consistent across studies [4,5,6]. The structure of the hippocampus comprises 26 subfields (13 left and 13 right) with distinct morphologies [7,8].

Increased right hippocampal volumes have been found in female responders compared to nonresponders after 8 weeks of fluoxetine treatment among MDD patients [9]. On the other hand, treatment with escitalopram did not result in a change the hippocampal volume [10]. Maller et al. [11] determined that a larger hippocampal tail volume was positively related to clinical remission between patients who did and did not undergo remission to antidepressant medications in volume analyses of 12 hippocampal subfields. Cao et al. [12] found that ECT-induced volume increases in the cornua Ammonis (CAs), the dentate gyrus layer (GCL), the molecular layer (ML) and the subiculum by using a segmentation pipeline. Thus, it has not been elucidated whether treatment with antidepressants affects the hippocampal volume, and changes in hippocampal volume after recovery from MDD remain unclear. It remains unclear if hippocampal volume is reduced in hippocampus in MDD or if volume increases in remission. Volume reduction may be due to disease severity, duration, or recurrent nature of MDD, or to the age of onset.

To the best of the authors’ knowledge, no previous reports have investigated the volume of hippocampal subfields after 6 months of continuous remission in MDD patients after treatment with antidepressants, therefore the aim of the present study was to investigate the changes in the volumes of the hippocampus and its subfields in MDD patients who responded to antidepressants and subsequently are in continuous remission.

## 2. Results

### 2.1. Participants

Eighteen participants responded to treatment with antidepressants, and none were relapsed within at least 6 months. We demonstrate the HAMD score trajectory of each case in Figure 1. The subfields of the hippocampus are shown in Figure 2a–c.

### 2.2. Volume of Whole-Hippocampus

No significant difference was found between the volume of whole hippocampus at baseline and six months after treatment initiation (Figure 3a,b, Table 1a,b).

### 2.3. Volumes of Hippocampal Subfields

A significant volume increase was found at 6 months in the molecular layer of hippocampus, the GC-MLDG, the CA3 (Figure 3a). We did not find a significant change in any other subfield volume at right (Figure 3b).

### 2.4. Change in the HAMD Score and in the Volume of the DG and CA3 Region

There was no correlation between the change in the HAMD score and the volume increases in the molecular layer of the left hippocampus, the left GC-DG, the left CA3 regions (Figure 4).

## 3. Discussion

We recruited first-episode, drug-naïve MDD patients and subsequently followed them for at least 6 months. All participants had undergone baseline MRI before starting any treatments, including pharmacotherapy. The volumes of the left cornu Ammonis (CA) 3, the left CG-DG and the whole hippocampus were increased in the MDD patients who responded to antidepressants and were subsequently in continuous remission. However, the volumes of the subfields and the whole hippocampus were not correlated to the HAMD score. The reason for this lack of correlation remains unknown. The weakest point of the present study is the lack of the normal controls and MDD patients who were not in continuous remission. Future studies should consider investigating the correlation using only MDD patients who did not undergo remission due to the negative results.

Brain imaging studies of the hippocampus in patients and stress-induced animal models with either depression or anxiety disorders indicate a remarkable reduction in hippocampal region volume and the number of dendritic spines [13,14]. Potentially underlying these structural anomalies, chronic stress has been shown to have detrimental effects on hippocampal neurogenesis and neuroplasticity in these individuals [15,16], consequently leading to cognitive and emotional symptoms of depression and anxiety. The hippocampus is not a uniform structure and consists of several subfields, such as CA (1–4) and the DG, which includes a GCL and an ML. It is known that cellular and molecular mechanisms associated with MDD may be localized to specific hippocampal subfields. Thus, it is necessary to investigate the link between the in vivo hippocampal subfield volumes and MDD. Hippocampal subfields CA (1–3) are reduced by experimental stress in animal studies [17], which is consistent with human behavioral findings of the preferential impact of early-life maltreatment stress, particularly in high-risk patients [18]. The sub-granular zone of the dentate gyrus is only source of neurogenesis in the hippocampus [19]. Thus, the finding of an increase in the volume of these regions observed after remission and subsequent recovery in MDD patients is not a contradiction. The treatment with fluoxetine, a selective serotonin reuptake inhibitor promotes the hippocampal neurogenesis in rodents [20]. Based on the postmortem study, dentate granule cell number and dentate gyrus size in medicated patients with depression are larger than those in nonmedicated patient [21]. SSRIs, lithium, and electroconvulsive therapy produce larger increases in hippocampus volume in treated depressed patients than in nontreated patients [22,23]. Huang et al. reported hippocampal subfields has revealed larger dentate gyri in medicated depressed patients [24]. Although the mechanisms of the neurogenesis remain unknown, glycogen synthase kinase-3β/β-catenin signaling might be involved in the mechanism of how antidepressants might influence hippocampal neurogenesis [25]. Recently, it has been reported that Tropomyosin receptor kinase B-dependent neuronal differentiation is involved in the sustained antidepressant effects of ketamine, which has potent antidepressive efficacy [26]. Nevertheless, there is no adequate evidence to establish that hippocampal neurogenesis is necessary for the antidepressant’s efficacy [27,28]. The volumes of the bilateral CA1, bilateral CA4, left CA2/3, bilateral GCL, and bilateral ML were decreased in medicated MDD patients compared with healthy subjects [29]. The volumes of GC-DG and CA (1–3) were also decreased in unmedicated MDD patients compared with healthy controls [24]. On the other hand, Brown et al. [30] reported that there were no significant differences between MDD patients and healthy subjects in hippocampal subfield volume. Recently, we also reported that the volumes of all hippocampal subfields did not significantly differ between MDD patients and healthy controls [31]. Taken together, it remains controversial whether a significant difference exists in hippocampal subfield volume between MDD patients and healthy subjects.

FreeSurfer v6.0 was used for image processing. For all scans, “recon-all” and Longitudinal Processing [32,33] were performed first. After Longitudinal Processing, brain structure volumes were calculated based on Aseg segmentation. In addition, we calculated hippocampal subfield volumes using longitudinal segmentation [24,34]. The motivation behind the use of Longitudinal Processing was to generate topologically equivalent surface meshes for any volumes under comparison. The surface mesh generated from the unbiased within-subject template was used for repositioning of the surface mesh relative to each volume. The repositioning procedure provided surfaces with the same geometry. Therefore, longitudinal processing may provide a more accurate estimate of differences [32,33,35].

Several limitations exist in the present study. First, this study has a small sample size, which was heterogeneous. Second, pharmacological treatment regulated. Third, we could not compare the difference in volume change between the two groups referring to single episode vs. recurrent/multiple episodes in the present study because of the few subjects in recurrent/multiple episodes. Fourth, we included neither a control group nor a group that did not undergo remission. Therefore, we should perform further studies considering the above points. We are now performing new project comparing the hippocampal volume of MDD patients in first episode and recurrent episodes treated with SSRIs and SNRIs with a larger sample.

In conclusion, an increase in the volume of CA3 and CG-DC of the only left, but not right hippocampus was observed in drug-naive MDD patients who subsequently underwent remission for at least 6 months with antidepressant use.

## 4. Material and Methods

### 4.1. Ethics Statement

The study protocol was approved by the Ethics Committee (approval number: H25-13, 8 May 2013) of the University of Occupational and Environmental Health, Japan. Written informed consent was obtained from all subjects who participated in this study. Informed consent was obtained from each patient in accordance with the Declaration of Helsinki.

### 4.2. Participants

The patients with MDD were recruited from the University hospital of University of Occupational and Environmental Health, Japan. The subjects in the present study partially overlapped with those in our recently published study. Eighteen patients with major depressive disorder were additionally enrolled in the present study. The MDD patients were recruited from March 2009 to January 2017. All patients were diagnosed by using the full Structured Clinical Interview from the Diagnostic and Statistical Manual for Mental Disorders, Fourth Edition, Text revision, Research Version. The severity of the depressive state was evaluated using the 17-item Hamilton Rating Scale for Depression (HAMD). Patients who met the following criteria were enrolled in the study: (a) a diagnosis of MDD, (b) a HAMD score of ≥14, (c) drug-naïveté MDD, and (d) response to antidepressant treatment (a 50% or more reduction in the HAMD score at week 8 from week 0). The exclusion criteria were as follows: (a) a history of neurological disease and/or the presence of psychiatric disorders on either Axis I (schizophrenia, other affective disorders, etc.) or Axis II (personality disorders, mental retardation, etc.), (b) the presence of comorbid substance use disorders, and (c) lack of a second MRI. Eighteen right-handed drug-naïve patients with MDD were included in this study. The clinical and demographic characteristics of the patients are summarized in Table 2. Patients took one of several classes of antidepressants, including selective serotonin reuptake inhibitors, i.e., escitalopram (*n* = 2, maximum dose: 20 mg/day), sertraline (*n* = 3, maximum dose: 150 mg/day), paroxetine (*n* = 5, maximum dose: 40 mg/day), and fluvoxamine (*n* = 3, maximum dose: 200 mg/day), and serotonin and norepinephrine reuptake inhibitors, i.e., duloxetine (*n* = 3, maximum dose: 60 mg/day), and mirtazapine (*n* = 2, maximum dose: 45 mg/day) at least 6 months or more.

All participants underwent T1-weighted structural MRI and were treated with antidepressants for 8 weeks. We defined patients whose HAM-D scores were <8 points or less as being in remission. We compared the volume of the hippocampus, including its subfields, in patients in remission at baseline to the volume at 6 months (Table 2).

### 4.3. MRI Acquisition

All participants underwent T1-weighted structural MRI at baseline and at 6 months after starting antidepressant therapy. The MRI data were obtained using a 3 T MR System (Signa EXCITE 3T; GE Healthcare, Wankesha, WI, USA) with an 8-channel brain phased-array coil. The original T1 images were acquired using a 3D fast-spoiled gradient recalled acquisition in the steady state. The acquisition parameters were as follows: repetition time in ms/echo time in ms/inversion time in ms = 10/4.1/700; flip angle = 10; field-of-view = 24 cm; section thickness = 1.2 mm, and resolution = 0.9 × 0.9 × 1.2 mm^3^. All the images were corrected for image distortion resulting from gradient nonlinearity using the Grad Warp software program and from intensity inhomogeneity using the “N3” function [36].

### 4.4. Image Processing

FreeSurfer v6.0 (http://surfer.nmr.mgh.harvard.edu/) was used for image processing. For all scans, “recon-all” and Longitudinal Processing [32,33] were performed first. For each subject, templates for one scanning session were generated from images acquired at baseline and 6 months later. After Longitudinal Processing, brain structure volumes were calculated based on Aseg segmentation. In addition, we calculated hippocampal subfield volumes using longitudinal segmentation [24,34]. The motivation behind the use of Longitudinal Processing was to generate topologically equivalent surface meshes for any volumes under comparison. The surface mesh generated from the unbiased, within-subject template was used for repositioning of the surface mesh relative to each volume. The repositioning procedure provided surfaces with the same geometry. Therefore, Longitudinal Processing may provide a more accurate estimate of differences among subjects [32,33,35]. Although the results provided by Longitudinal Processing may not be generalized to those provided with a conventional stream, the same differences can be detected by pooling measurements across a large population to average the processing bias [35].

### 4.5. Statistical Analysis

We used paired t-rests to compare the change in hippocampal subfield volume between baseline and after 6 months. We calculated Pearson’s correlation coefficients between the change in the HAMD score and the change in hippocampal subfield volume. We regarded values of *p* < 0.05 (two tailed) as a statistically significant difference.

## Figures and Tables

**Figure 1 ijms-21-03032-f001:**
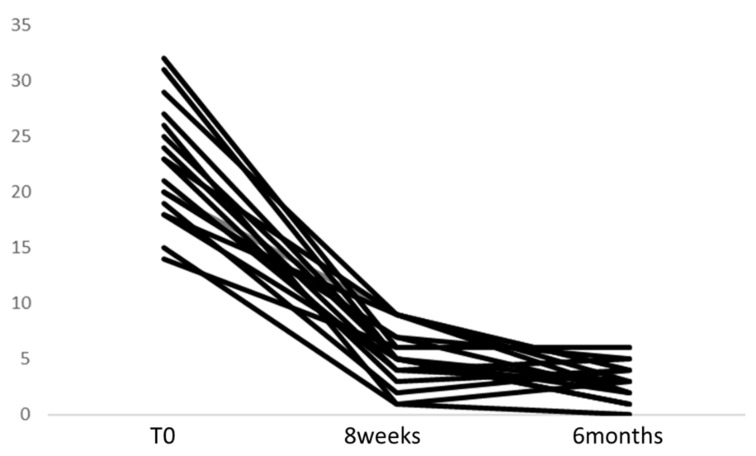
The HAMD score trajectory of each MDD case. Each line presents the change the HAMD score at baseline, 8 weeks, and 6 months after starting antidepressants.

**Figure 2 ijms-21-03032-f002:**
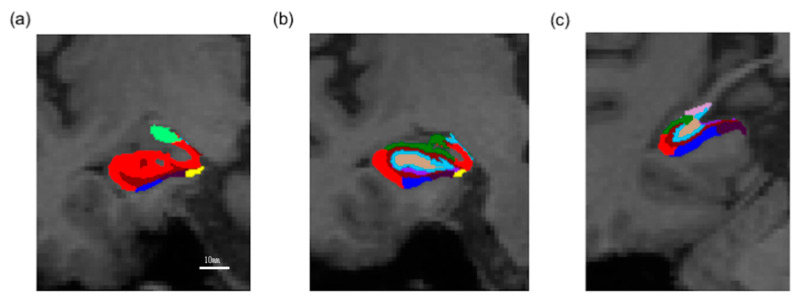
Representative subdivision of the hippocampal subfields. The mask of each region was overlaid on coronal T1-weighted images from anterior (**a**), middle (**b**), to posterior (**c**). Color classification: parasubiculum = yellow; presubiculum = black; subiculum = blue; cornu Ammonis (CA) 1 = red; CA3 = dark green; CA4 = brown; granule cell layer of the dentate gyrus (GC-DG) = sky blue; hippocampus-amygdala transition area (HATA) = green; fimbria = purple; molecular layer of the hippocampus (HP) = dark brown; hippocampal fissure = dark purple; hippocampal tail = gray.

**Figure 3 ijms-21-03032-f003:**
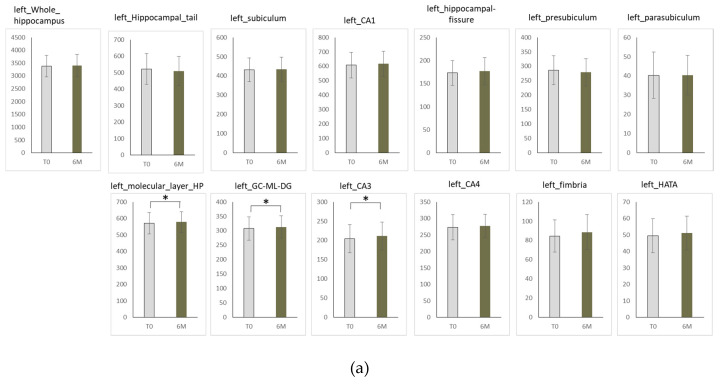
Subfield volume in hippocampus at left (**a**) and right (**b**). T0; baseline, 6M; after 6 months, Y-axis shows volume in hippocampus subfields (mm^3^).Vertical bar means standard error (SEM).

**Figure 4 ijms-21-03032-f004:**
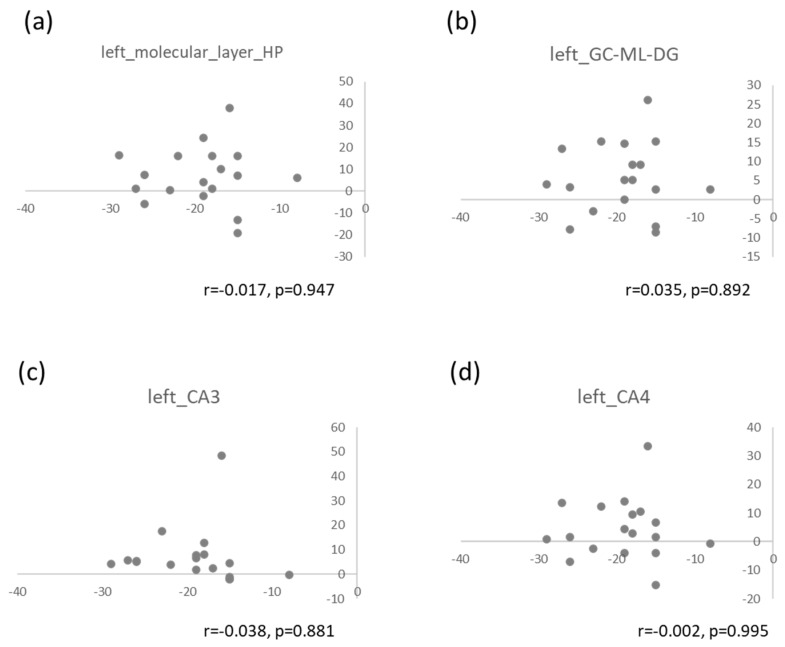
Difference in the HAMD score and in the volume of the left molecular layer (**a**), left GC-ML-DG (**b**), left CA3 (**c**), and left CA4 (**d**). X-axis shows the difference of the HAMD score baseline and 6 months; Y-axis shows the difference of volume (mm^3^) at baseline and 6 months.

**Table 1 ijms-21-03032-t001:** Volumes of Hippocampal Subfields at left (**a**), and right (**b**).

(a)				(b)			
	T0	6M			T0	6M	
n	18	18		n	18	18	
	Mean ± SD (mm^3^)	Mean ± SD (mm^3^)	*p*-value		Mean ± SD (mm^3^)	Mean ± SD (mm^3^)	*p*-value
left_Hippocampal_tail	521.9 ± 93.2	509.7 ± 88.8	0.063	right_Hippocampal_tail	565.2 ± 94.9	560.3 ± 98.6	0.428
left_subiculum	432.0 ± 61.8	434.6 ± 62.7	0.36	right_subiculum	444.9 ± 65.9	445.4 ± 68.2	0.837
left_CA1	608.6 ± 88.6	617.0 ± 86.6	0.074	right_CA1	653.7 ± 92.9	652.8 ± 97.4	0.808
left_hippocampal-fissure	173.1 ± 27.0	176.9 ± 29.8	0.358	right_hippocampal-fissure	199.9 ± 44.7	201.9 ± 39.4	0.711
left_presubiculum	286.0 ± 50.2	279.3 ± 46.7	0.088	right_presubiculum	263.2 ± 41.7	259.2 ± 43.7	0.314
left_parasubiculum	40.3 ± 12.1	40.3 ± 10.3	0.984	right_parasubiculum	43.2 ± 8.8	42.7 ± 9.5	0.505
left_molecular_layer_HP	571.1 ± 64.3	578.0 ± 63.7	0.046	right_molecular_layer_HP	604.8 ± 64.8	608.2 ± 71.3	0.347
left_GC-ML-DG	307.6 ± 40.3	313.2 ± 38.7	0.021	right_GC-ML-DG	338.2 ± 37.8	340.1 ± 40.4	0.495
left_CA3	204.3 ± 36.4	211.5 ± 36.1	0.016	right_CA3	232.1 ± 33.2	235.3 ± 35.5	0.059
left_CA4	273.1 ± 37.8	277.4 ± 34.6	0.1	right_CA4	299.0 ± 34.9	301.3 ± 37.0	0.34
left_fimbria	84.6 ± 16.9	88.5 ± 18.1	0.135	right_fimbria	78.9 ± 23.1	79.2 ± 23.5	0.892
left_HATA	49.5 ± 10.4	51.1 ± 10.2	0.099	right_HATA	55.2 ± 9.5	55.8 ± 9.5	0.308
left_Whole_hippocampus	3379.0 ± 381.1	3400.5 ± 376.8	0.131	right_Whole_hippocampus	3578.3 ± 421.7	3580.2 ± 444.0	0.916

**Table 2 ijms-21-03032-t002:** The characteristics of patients, the HAMD score, and dose of antidepressants.

N	18
Age	44.4 ± 13.8
Gender(Male/Female)	10/8
Episode(First/Recurrent)	12/6
HAM-D T0	22.5 ± 4.99
HAM-D 8weeks	5.4 ± 2.6
HAM-D 6months	3.2 ± 1.7
dose at 8 weeks (imipramine equivalent)	188 ± 105.2

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
