# Peer review of "Structural Changes in Hippocampal Subfields in Patients with Continuous Remission of Drug-Naive Major Depressive Disorder"

_ijms, 2020, doi:10.3390/ijms21093032_

Round 1

Reviewer 1 Report

Katsuki and coworkers have used MRI to measure the volume of different hippocampal fields in 18 drug-naïve MDD patients before treatment and at recovery 6 months later.  They report modest increases in left CA3 and in granule cell layer of the dentate gyrus.  No correlation between treatment response and volume changes were found.  No controls or patients that failed to recover were imaged.  Although there are several prior studies where hippocampal volumes were investigated in depressed patients, the issue is controversial and therefore novel studies are needed.   Overall, the value of this study is compromised by a small sample size and lack of proper controls, but the experimental design where a patient is compared to him/herself before and after treatment allows some conclusions to be drawn.  However there are several issues in the manuscript that potentially severely influence the conclusions drawn.

  1. The most important issue is the statistics used.The authors have used paired t-test to test for significances.  They have measured 13 hippocampal regions bilaterally hence, they have performed 26 t-tests.  When significance level is set at 0.05, one comparison in 20 is expected to be significant by chance. The authors should, for example, use a repeated Two-way ANOVA (factors: time point and brain region) combined with a multiple comparison test as post-hoc (such as Sidak or Holm-Sidak) to compare  before x after treatment in each individual brain area. This way the 26 comparison will be corrected for multiple comparisons. It is likely that after this comparison, none of the currently significant findings remain significant, however, the current method without any correction for multiple testing cannot be approved.
  2. Line 166 says that 20 patients are included, but results are given only on 18.Why?
  3. Figure 1: Please indicate what a-c indicate.
  4. Figure 4: what is shown on x and y axis, please add to the figure.
  5. Hippocampus has distinct functions along its antero-posterior axis (dorso-vertral in rodents), with anterior being more involved in emotional and posterior more in cognitive functions.I am wondering whether the data of the authors can reveal something along this antero-posterior axis?
  6. Line 116: it is unclear what the authors mean with “ML that continuously crosses the adjacent subiculum …”.If the authors mean to indicate that ML and subiculum are continuous, this is incorrect they are separated by meninges and other stuctures, although they may appear continuous in low resolution.
  7. Line 121: Neurogenesis in hippocampus is confined to the GC layer of the DG, there is no neurogenesis in CA3.However the projections of the newly born granule neurons do project to CA3, so neurogenesis can theoretically have a small impact on the volume of the CA3 area. Please correct.

Author Response

Reviewer 1

  1. The most important issue is the statistics used. The authors have used paired t-test to test for significances.  They have measured 13 hippocampal regions bilaterally hence, they have performed 26 t-tests.  When significance level is set at 0.05, one comparison in 20 is expected to be significant by chance. The authors should, for example, use a repeated Two-way ANOVA (factors: time point and brain region) combined with a multiple comparison test as post-hoc (such as Sidak or Holm-Sidak) to compare  before x after treatment in each individual brain area. This way the 26 comparison will be corrected for multiple comparisons. It is likely that after this comparison, none of the currently significant findings remain significant, however, the current method without any correction for multiple testing cannot be approved.

Our reply

We appreciate your comment. As you mentioned, previous neuroimaging study revealed that the effects of sex, age, and total brain volume (or head size) should be adjusted for between-subject comparisons (Barnes et al., 2010). However, we compared the same subjects before and after treatment. In this case, it is controversial whether these valuables should be adjusted.  For instance, a previous study also used paired t-test to evaluate the hippocampal volume change after the electroconvulsive study (Van Den Bossche et al., 2019).   

Barnes, J., Ridgway, G.R., Bartlett, J., Henley, S.M., Lehmann, M., Hobbs, N., Clarkson, M.J., MacManus, D.G., Ourselin, S., Fox, N.C., 2010. Head size, age and gender adjustment in MRI studies: a necessary nuisance? Neuroimage 53, 1244-1255.

Van Den Bossche, M.J.A., Emsell, L., Dols, A., Vansteelandt, K., De Winter, F.L., Van den Stock, J., Sienaert, P., Stek, M.L., Bouckaert, F., Vandenbulcke, M., 2019. Hippocampal volume change following ECT is mediated by rs699947 in the promotor region of VEGF. Translational psychiatry 9, 191.

  1. Line 166 says that 20 patients are included, but results are given only on 18.Why?

Our reply

The number of the subjects was eighteen. We corrected the number from 20 to 18 in the manuscript.

  1. Figure 1: Please indicate what a-c indicate.

Our reply

We added the sentence in Figure 1 legend as follows.

The mask of each region was overlaid on coronal T1-weighted images from anterior (a) to posterior (c).

  1. Figure 4: what is shown on x and y axis, please add to the figure.
  2. Hippocampus has distinct functions along its antero-posterior axis (dorso-vertral in rodents), with anterior being more involved in emotional and posterior more in cognitive functions. I am wondering whether the data of the authors can reveal something along this antero-posterior axis?

Our reply

We cannot demonstrate any data about antero-posterior axis. We can only give data for the hippocampal subfields.

  1. Line 116: it is unclear what the authors mean with “ML that continuously crosses the adjacent subiculum …”.If the authors mean to indicate that ML and subiculum are continuous, this is incorrect they are separated by meninges and other structures, although they may appear continuous in low resolution.

Our reply

We changed the phrase as follows because of it was imprecisely.

The hippocampus is not a uniform structure and consists of several subfields, such as CA (1–4) and the DG, which includes a GCL and an ML.

  1. Line 121: Neurogenesis in hippocampus is confined to the GC layer of the DG, there is no neurogenesis in CA3.However the projections of the newly born granule neurons do project to CA3, so neurogenesis can theoretically have a small impact on the volume of the CA3 area. Please correct.

Our reply

Thank you so much for your criticism. We corrected your suggestion as follows.

The GC-DG is considered the most active neurogenesis subfields in the hippocampus [19].

Reviewer 2

The authors examined changes in hippocampal volume and its subfields in patients with major depressive disorder who responded to antidepressants and were in continuous remission. A considerable strength of the study was the longitudinal design whereby hippocampal volume was measured at time 0 and again at 6 months. There was a substantial reduction in the score on the Hamilton Depression Rating Scale after 8 weeks of antidepressant drug treatment and this persisted to 6 months. Despite these strengths, there are significant weaknesses in the methodology and description of the results. Finally, except for a passing reference to neurogenesis (Line 121), no mechanisms are proposed for the reported results. What cellular features, beyond the reference to neurogenesis, may underlie the recrudescence of the hippocampus after successful treatment, and how do animal studies contribute to understanding the results?

Our reply

 The effects of treatment with fluoxetine, a selective serotonin reuptake inhibitor promotes the hippocampal neurogenesis in rodents [T.D.Perera,J.D.Coplan,S.H.Lisanbyetal.,“Antidepressant induced neurogenesis in the hippocampus of adult nonhuman primates,” Journal of Neuroscience,vol.27,no.18,pp.4894–4901, 2007.]. Based on the postmortem study, dentate granule cell number and dentate gyrus size in medicated patients with depression are larger than those in nonmedicated patient [M. Boldrini, T.H. Butt, A.N. Santiagoetal.,“Benzodiazepines and the potential trophic effect of antidepressants on dentate gyrus cells in mood disorders,” International Journal of Neuropsychopharmacology,vol.17,no.12,pp.1923–1933,2014.]. Moreover, SSRIs, lithium, and electroconvulsive therapy produce larger increases in hippocampus volume in treated depressed patients than in nontreated patients [B. Hallahan, J. Newell, J. C. Soares et al., “Structural magnetic resonance imaging in bipolar disorder: an international collaborative mega-analysis of individual adult patient data,” BiologicalPsychiatry,vol.69,no.4,pp.326–335,2011., I. Tendolkar, M. van Beek, I. van Oostrom et al., “Electroconvulsive therapy increases hippocampal and amygdala volume in therapy refractory depression: A Longitudinal Pilot Study,” Psychiatry Research—Neuroimaging,vol.214,no.3,pp.197–203, 2013.]. Huang et al. reported that hippocampal subfields has revealed larger dentate gyri in medicated depressed patients [Y. Huang, N.J. Coupland, R.M. Lebel et al., “Structural changes in hippocampal subfields in major depressive disorder: a high field magnetic resonance imaging study,” Biological Psychiatry, vol.74,no.1,pp.62–68, 2013.]. Although the mechanisms of the neurogenesis remain unknown, glycogen synthase kinase-3 beta/β-catenin signaling might be involved in the mechanism of how antidepressants might influence hippocampal neurogenesis (Hui J, Zhang J, Kim H, et al. Fluoxetine regulates neurogenesis in vitro through modulation of GSK-3β/β-catenin signaling. Int J Neuropsychopharmacol. 2014;18(5):pyu099. Published 2014 Dec 7. doi:10.1093/ijnp/pyu099). Recently, It has been reported that Tropomyosin receptor kinase B-dependent neuronal differentiation is involved in the sustained antidepressant effects of ketamine, which has potent antidepressive efficacy (Ma Z, Zang T, Birnbaum SG, et al. TrkB dependent adult hippocampal progenitor differentiation mediates sustained ketamine antidepressant response. Nat Commun. 2017;8(1):1668. Published 2017 Nov 21. doi:10.1038/s41467-017-01709-8). Nevertheless, it is no adequate evidence to establish that hippocampal neurogenesis is necessary for the antidepressant’s efficacy (Sahay A, Hen R. Adult hippocampal neurogenesis in depression. Nat Neurosci. 2007;10(9):1110–1115. doi:10.1038/nn1969. Tunc-Ozcan E, Peng CY, Zhu Y, Dunlop SR, Contractor A, Kessler JA. Activating newborn neurons suppresses depression and anxiety-like behaviors. Nat Commun. 2019;10(1):3768. Published 2019 Aug 21. doi:10.1038/s41467-019-11641-8).

Methods – There is uncertainty as to the number of participants. 18 participants responded (line 67); 20 right-handed drug-naïve patients with MDD were included in the study (line 166); Figure 2 appears to only show 17 lines. There is no information on the dosages and durations of treatment with the various antidepressant medications.

Our reply

 Each antidepressant was prescribed full dose permitted in Japan at least six months or more. We added the information in the text as follows.

Patients had taken several classes of antidepressants, including selective serotonin reuptake inhibitors, i.e., escitalopram (n=2, maximum dose: 20mg/day), sertraline (n=3, maximum dose: 150mg/day), paroxetine (n=5, maximum dose: 40mg/day), and fluvoxamine (n=3, maximum dose: 200mg/day), and serotonin and norepinephrine reuptake inhibitors, i.e., duloxetine (n=3, maximum dose: 60mg/day), and mirtazapine (n=3, maximum dose: 45mg/day) at least 6 months or more.

Results –

  • Figure 1 – The hippocampal images need to be significantly enlarged with enhanced resolution so that the hippocampal subfields can be viewed.

Our reply

We added the new Figure enlarged the hippocampal images. We enlarged the hippocampal image with enhanced resolution (300 dpi).

  • While lines 80-81 note “No significant difference was found between the whole brain volumes at baseline and six months after treatment initiation (Figure 3a and b)”, no data are included on whole brain volumes. These data should be included.

Our reply

We apologize our mistake. We changed the sub-title of 2.2 and sentence as follows. We meant to mention the volume of whole hippocampus, not whole brain.

2.2. Volume of Whole-hippocampus

No significant difference was found between the volume of whole hippocampus at baseline and six months after treatment initiation (Figure 3a and b).

  • A Table should be included noting mean, N and standard deviations related to subfield volumes. For comparisons of Time 0 vs. 6 months data, include all statistical data.

Our reply

We made the new table considering your suggestion. (Table1)。

  • It is not clear if there are differences in the volume responses between patients taking an SSRI vs those taking the other medications?

Our reply

We are now writing new paper comparing the hippocampal volume MDD patients treated with SSRIs or SNRIs with a larger sample. We inserted the following sentence in the text as limitations in the study.

Several limitations exist in the present study. First, this study has a small sample size, which was heterogeneous. Second, pharmacological treatment regulated. thus, we could not elucidate which type of antidepressant most influence the hippocampal volume. Third, Fourth, we included neither a control group nor a group that did not undergo remission. Therefore, we should perform further studies considering the above points. We are now performing new project comparing the hippocampal volume of MDD patients in first episode and recurrent episodes treated with SSRIs and SNRIs with a larger sample.

  • Figure 1 legend notes the granule cell layer of the dentate gyrus (GC-DG)(Line 74). The molecular layer of the hippocampus (HP) is also noted in Figure 1 legend (line 75). However, Figures 3 (a) and (b) note GC-ML-DG, an abbreviation not explained or used elsewhere. See also Figure 4 (b).
  • (Lines 83-84) “A significant volume increase was found at 6 months in the molecular layer of the left hippocampus, the left GC-DG, the left CA3 and the left CA4 (Figure 3a and b).” There is confusion between this sentence and the Figure 3 (a) use of left_GC-ML-DG.

Our reply

We corrected the point.

The Figure legend in Line 86 is confusing: “Figure 3. and subfield volume in hippocampus at left (a) and right (b).”

Our reply

We corrected the point.

It would be of interest to note if there is a correlation between the difference in Ham-D score between time 0 and Week 8 vs. the difference in hippocampal subregional volume at time 0 and 6 months? In other words, does the Ham-D improvement at 8 weeks predict the hippocampal volume change at 6 months?

Our reply

There were not significant correlations between the difference in Ham-D score between time 0 and Week 8 vs. the difference in hippocampal any subregional volume at time 0 and 6 months. We added the result as follows in the text..

There was no correlation between the change in the HAMD score and the volume increases in the molecular layer of the left hippocampus, the left GC-DG, the left CA3 and the left CA4 regions

  • Were these depressive episodes the first such episodes in the lives of these patients? Is there a difference in volume response between those with first-episode depression and those with recurrent depression?

Our reply

Several limitations exist in the present study. First, this study has a small sample size, which was heterogeneous. Second, pharmacological treatment regulated. thus, we could not elucidate which type of antidepressant most influence the hippocampal volume. Third, Fourth, we included neither a control group nor a group that did not undergo remission. Therefore, we should perform further studies considering the above points. We are now performing new project comparing the hippocampal volume of MDD patients in first episode and recurrent episodes treated with SSRIs and SNRIs with a larger sample.

  • There is no statistical evaluation of or control for potential effects of sex, age or total brain volume (Malykhin et al., 2008; Adler et al., 2018; Schmidt et al., 2018; Stein et al., 2012), nor is there a statistical correction for multiple comparisons when comparing hippocampal subregion volume at time 0 vs. 6 months. Simple paired t-tests are not sufficient.

Our reply

We appreciate your comment. As you mentioned, previous neuroimaging study revealed that the effects of sex, age, and total brain volume (or head size) should be adjusted for between-subject comparisons (Barnes et al., 2010). However, we compared the same subjects before and after treatment. In this case, it is controversial whether these valuables should be adjusted.  For instance, a previous study also used paired t-test to evaluate the hippocampal volume change after the electroconvulsive study (Van Den Bossche et al., 2019).  

Barnes, J., Ridgway, G.R., Bartlett, J., Henley, S.M., Lehmann, M., Hobbs, N., Clarkson, M.J., MacManus, D.G., Ourselin, S., Fox, N.C., 2010. Head size, age and gender adjustment in MRI studies: a necessary nuisance? Neuroimage 53, 1244-1255.

Van Den Bossche, M.J.A., Emsell, L., Dols, A., Vansteelandt, K., De Winter, F.L., Van den Stock, J., Sienaert, P., Stek, M.L., Bouckaert, F., Vandenbulcke, M., 2019. Hippocampal volume change following ECT is mediated by rs699947 in the promotor region of VEGF. Translational psychiatry 9, 191.

  • Pearson correlation coefficients and significance values should be included on the graphs in Figure 4.

Our reply

We added the Pearson correlation coefficients and significance values.

Discussion –

  • Since the patients were drug-naïve (line 97), on line 99, it should read “Patients took (not had taken) several classes of …

Our reply

We corrected the point.

  • Lines 102-104: The sentence “The volumes of the left cornu Ammonis (CA) 3, the left CG-DG and the whole hippocampus were increased in the MDD patients who responded to antidepressants and were subsequently in continuous remission” suggests that some patients did not respond. Who did not respond and were their hippocampal volumes also recorded? Are these the 2 patients noted above in my comment (20 included, 18 responded)?

Our reply

The all participants were eighteen. We corrected the point in the text.

  • Under Limitations (lines 142-143), “Second, pharmacological treatment, including the duration of administration, was not regulated.” Other than the different medications that were administered, what precisely does ‘not regulated’ mean? Lines 170-172: “All participants underwent T1-weighted structural MRI and were treated with antidepressants for 8 weeks.” How was ‘duration’ not regulated?

Our reply

All patients were started after the MRI scanning. The only a few days differed among the patients. We described as follows.

Several limitations exist in the present study. First, this study has a small sample size, which was heterogeneous. Second, pharmacological treatment regulated. thus, we could not elucidate which type of antidepressant most influence the hippocampal volume. Third, we included neither a control group nor a group that did not undergo remission. Therefore, we should perform further studies considering the above points. We are now performing new project comparing the hippocampal volume of MDD patients treated with SSRIs and SNRIs with a larger sample.

Reviewer 3

The current work examines hippocampal subfield volumes in a cohort of patients with 6 month remission to MDD. I have concerns about several aspects of the study, as detailed below.

Introduction:

The introduction is poorly referenced – needs work throughout to substantiate claims made.

The authors should moderate their wording on hippocampal volume reduction – several studies have not shown volume reductions in hippocampus in MDD or volume increases in remission (including the author’s present work). Volume reduction may be due to disease severity, duration or recurrent nature of MDD, age of onset etc.

Our reply

We inserted the following sentence in the text as follows.

It remains unclear hippocampal volume reduction in hippocampus in MDD or volume increases in remission. Volume reduction may be due to disease severity, duration or recurrent nature of MDD, age of onset also unknown.

Methods:

Insufficient information is provided regarding participants:

  • Where were they recruited from?

Our reply

The patients with MDD were recruited from the University hospital of University of Occupational and Environmental Health, Japan.

We added the point in the text as follows.

  • What was the antidepressant – no mention of this at all in methods or results? What was the dose and duration of treatment?

Our reply

We explained the point in the text.

  • What is their disease background, in terms of number of episodes, age of onset, etc?
  • At what stage were the participants recruited? They must have been recruited prior to baseline scan – are the authors only presenting a subset of their study for those who remitted, and remained remitted at 6 months? If so, this should be clearly stated.

Results:

Figures should be reordered so that they better match the flow of results:

  • Figure 1 should show the HAMD score trajectory while describing patient remissions as the first result;
  • Figure 2 should show hippocampal subfields
  • Figure 3 should show subfield volume reductions.

Our reply

Result 2.3 Whole brain analysis – this result is inconsistent with Figure 3, which does not show whole brain volumes, but rather hippocampal subfield volumes. If whole brain analysis has been performed, it should be presented in more detail and with an accompanying figure.

Our reply

Whole hippocampal, but not whole brain.  We could not investigate whole brain analysis.

Figure 3a – do error bars represent SD or SEM? p values should be presented alongside additional metrics in the results e.g. % increase. The plots do not seem to show a difference by eye.

Our reply

Vertical bar means standard error (SEM). We inserted p-value in the Table in the text.

Discussion:

This is the first mention of medications or first episode status of these patients. This should be clearly stated in the methods.

Our reply

The MDD patients were all drug-naïve, however including both first and recurrent episode. We corrected in the text.

The medication descriptions count n = 19 patients – did one take two medications, or one is not presented? This should be clearer, and in the methods.

Our reply

escitalopram (n=2, maximum dose: 20mg/day), sertraline (n=3, maximum dose: 150mg/day), paroxetine (n=5, maximum dose: 40mg/day), and fluvoxamine (n=3, maximum dose: 200mg/day), and serotonin and norepinephrine reuptake inhibitors, i.e., duloxetine (n=3, maximum dose: 60mg/day), and mirtazapine (n=2, maximum dose: 45mg/day)

The findings should be placed into context more clearly – why left subfield changes and not right?

Our reply

We changed conclusion more clearly as follows.

In conclusion, an increase in the volume of CA3, CA4 and CG-DC of the only light, but not right hippocampus was observed in drug-naive MDD patients who subsequently underwent remission for at least 6 months with antidepressant use.

Minor comments:

Abstract: “Hamilton Dating Scale” change “Dating” to Rating.

Our reply

We corrected the typo.

Would help the reader if X and Y axis was labelled on each figure, as well as stated in legend.

Our reply

We inserted the points in the legends.

Methods describe in text 20 patients, not 18.

Our reply

We corrected the number.

Reviewer 2 Report

The authors examined changes in hippocampal volume and its subfields in patients with major depressive disorder who responded to antidepressants and were in continuous remission. A considerable strength of the study was the longitudinal design whereby hippocampal volume was measured at time 0 and again at 6 months. There was a substantial reduction in the score on the Hamilton Depression Rating Scale after 8 weeks of antidepressant drug treatment and this persisted to 6 months. Despite these strengths, there are significant weaknesses in the methodology and description of the results. Finally, except for a passing reference to neurogenesis (Line 121), no mechanisms are proposed for the reported results. What cellular features, beyond the reference to neurogenesis, may underlie the recrudescence of the hippocampus after successful treatment, and how do animal studies contribute to understanding the results?

Methods – There is uncertainty as to the number of participants. 18 participants responded (line 67); 20 right-handed drug-naïve patients with MDD were included in the study (line 166); Figure 2 appears to only show 17 lines. There is no information on the dosages and durations of treatment with the various antidepressant medications.

Results –

  • Figure 1 – The hippocampal images need to be significantly enlarged with enhanced resolution so that the hippocampal subfields can be viewed.
  • While lines 80-81 note “No significant difference was found between the whole brain volumes at baseline and six months after treatment initiation (Figure 3a and b)”, no data are included on whole brain volumes. These data should be included.
  • A Table should be included noting mean, N and standard deviations related to subfield volumes. For comparisons of Time 0 vs. 6 month data, include all statistical data.
  • It is not clear if there are differences in the volume responses between patients taking an SSRI vs those taking the other medications?
  • Figure 1 legend notes the granule cell layer of the dentate gyrus (GC-DG)(Line 74). The molecular layer of the hippocampus (HP) is also noted in Figure 1 legend (line 75). However, Figures 3 (a) and (b) note GC-ML-DG, an abbreviation not explained or used elsewhere. See also Figure 4 (b).
  • (Lines 83-84) “A significant volume increase was found at 6 months in the molecular layer of the left hippocampus, the left GC-DG, the left CA3 and the left CA4 (Figure 3a and b).” There is confusion between this sentence and the Figure 3 (a) use of left_GC-ML-DG.
  • The Figure legend in Line 86 is confusing: “Figure 3. and subfield volume in hippocampus at left (a) and right (b).”
  • It would be of interest to note if there is a correlation between the difference in Ham-D score between time 0 and Week 8 vs. the difference in hippocampal subregional volume at time 0 and 6 months? In other words, does the Ham-D improvement at 8 weeks predict the hippocampal volume change at 6 months?
  • Were these depressive episodes the first such episodes in the lives of these patients? Is there a difference in volume response between those with first-episode depression and those with recurrent depression?
  • There is no statistical evaluation of or control for potential effects of sex, age or total brain volume (Malykhin et al., 2008; Adler et al., 2018; Schmidt et al., 2018; Stein et al., 2012), nor is there a statistical correction for multiple comparisons when comparing hippocampal subregion volume at time 0 vs. 6 months. Simple paired t-tests are not sufficient.
  • Pearson correlation coefficients and significance values should be included on the graphs in Figure 4.

Discussion –

  • Since the patients were drug-naïve (line 97), on line 99, it should read “Patients took (not had taken) several classes of …
  • Lines 102-104: The sentence “The volumes of the left cornu Ammonis (CA) 3, the left CG-DG and the whole hippocampus were increased in the MDD patients who responded to antidepressants and were subsequently in continuous remission” suggests that some patients did not respond. Who did not respond and were their hippocampal volumes also recorded? Are these the 2 patients noted above in my comment (20 included, 18 responded)?
  • Under Limitations (lines 142-143), “Second, pharmacological treatment, including the duration of administration, was not regulated.” Other than the different medications that were administered, what precisely does ‘not regulated’ mean? Lines 170-172: “All participants underwent T1-weighted structural MRI and were treated with antidepressants for 8 weeks.” How was ‘duration’ not regulated?

Author Response

Please see the attatched file.

Reviewer 3 Report

The current work examines hippocampal subfield volumes in a cohort of patients with 6 month remission to MDD. I have concerns about several aspects of the study, as detailed below.

Introduction:

The introduction is poorly referenced – needs work throughout to substantiate claims made.

The authors should moderate their wording on hippocampal volume reduction – several studies have not shown volume reductions in hippocampus in MDD or volume increases in remission (including the author’s present work). Volume reduction may be due to disease severity, duration or recurrent nature of MDD, age of onset etc.

Methods:

Insufficient information is provided regarding participants:

  • Where were they recruited from?
  • What was the antidepressant – no mention of this at all in methods or results? What was the dose and duration of treatment?
  • What is their disease background, in terms of number of episodes, age of onset, etc?
  • At what stage were the participants recruited? They must have been recruited prior to baseline scan – are the authors only presenting a subset of their study for those who remitted, and remained remitted at 6 months? If so, this should be clearly stated.

Results:

Figures should be reordered so that they better match the flow of results:

  • Figure 1 should show the HAMD score trajectory while describing patient remissions as the first result;
  • Figure 2 should show hippocampal subfields
  • Figure 3 should show subfield volume reductions.

Result 2.3 Whole brain analysis – this result is inconsistent with Figure 3, which does not show whole brain volumes, but rather hippocampal subfield volumes. If whole brain analysis has been performed, it should be presented in more detail and with an accompanying figure.

Figure 3a – do error bars represent SD or SEM? p values should be presented alongside additional metrics in the results e.g. % increase. The plots do not seem to show a difference by eye.

Discussion:

This is the first mention of medications or first episode status of these patients. This should be clearly stated in the methods.

The medication descriptions count n = 19 patients – did one take two medications, or one is not presented? This should be clearer, and in the methods.

The findings should be placed into context more clearly – why left subfield changes and not right?

Minor comments:

Abstract: “Hamilton Dating Scale” change “Dating” to Rating.

Would help the reader if X and Y axis was labelled on each figure, as well as stated in legend.

Methods describe in text 20 patients, not 18.

Author Response

Please see the attatched file.

Round 2

Reviewer 2 Report

Line 21 – “… scores on the 17-items of Hamilton Dating Scale for Depression (HAMD)”.

Change Dating to Rating.

Lines 47-48 – “MDD produces a volume reduction in the hippocampus compared with 47 healthy controls, a finding that has been relatively consistent across studies.”

“produces” should be changed to “is associated with”, since the hippocampus may already be smaller prior to the onset of depression.

Lines 59–61 - “It remains unclear << if >> hippocampal volume << is reduced >> in MDD or << if >> volume increases in remission. Volume reduction may be due to disease severity, duration or recurrent nature of MDD, << or to the >> age of onset. (remove ‘also unknown’). (Add/edit where << >> are found)

Lines 88-89 – “A significant volume increase was found at 6 months in the molecular layer of hippocampus, 88 the GC-MLDG, the CA3 and the CA4 at left (Figure 3).”

There is no statistically significant proof in Table 1a that the volume of left CA4 differs between time 0 and 6 months. CA4 should be removed from this sentence.

Lines 97-98- “There was no correlation between the change in the HAMD score and the volume increases in the molecular layer of the left hippocampus, the left GC-DG, the left CA3 and the left CA4 regions (Figure 4).”

There was no volume increase in left CA4 so it should be removed from this sentence.

Lines 116-117- “Figure 4. Difference in the HAMD score and in the volume of the left molecular layer (a), left CG-ML-DG(b),…”

Typo: Change left CG-ML-DG to “left GC-ML-DG(b)”

Line 122- “Patients not had taken several classes of antidepressants,…”

Change to: “Patients took one of several classes of antidepressants, …”

Lines 122-127- “Patients took one of several classes of antidepressants, including 122 selective serotonin reuptake inhibitors, i.e., escitalopram (n=2, maximum dose: 20mg/day), sertraline 123 (n=3, maximum dose: 150mg/day), paroxetine (n=5, maximum dose: 40mg/day), and fluvoxamine 124 (n=3, maximum dose: 200mg/day), and serotonin and norepinephrine reuptake inhibitors, i.e., 125 duloxetine (n=3, maximum dose: 60mg/day), and mirtazapine (n=2, maximum dose: 45mg/day) at 126 least 6 months or more.”

This sentence should be moved to Materials and Methods 4.2 after Table 2.

Lines 144-145- “The GC-DG is considered the most active neurogenesis subfields in the hippocampus [19].”

Cited reference 19 notes: “Immature neurons, once generated by transient-amplifying progenitor cells located in the sub-granular zone of the adult DG, …” The subgranular zone of the dentate gyrus is the ONLY source of neurogenesis in the hippocampus. Please reword Line 144 appropriately.

Lines 158-161- “The volumes of the bilateral CA1, bilateral CA4, left CA2/3, bilateral GCL, and bilateral ML were reduced in medicated MDD patients compared with healthy subjects [29]. The volumes of GC-DG and CA (1-3) were also reduced in unmedicated MDD patients compared with healthy controls [24].”

It is incorrect to say that volumes were “reduced” since these studies were cross-sectional. Only if there was a ‘before’ measurement followed by an ‘after’ measurement can one say volume was reduced or increased. Rather, change reduced to “decreased” in these 2 sentences.

Lines 175-176- “Second, pharmacological treatment regulated. thus, we could not elucidate which type of antidepressant most influence the hippocampal volume.”

The syntax of this sentence should be corrected. It is incomplete and disjointed as written. Explain how antidepressant treatment is a limitation?

Lines 176-178- “Third, we could not compare the difference in volume change between the two groups in the present study because of the few subjects in recurrent episodes.”

The meaning of this sentence is unclear. Are you referring to single episode vs. recurrent/multiple? If so, please be more explicit.

Line 182- “In conclusion, an increase in the volume of CA3, CA4 and CG-DC of the only left,…”

There was no statistically significant increase in volume of CA4. CA4 should be removed from this sentence.

Author Response

To the reviewer #2

Thank you so much for your helpful comments. We corrected all points according to your suggestion.

We hope the re-revised manuscript is suitable for publication.

Best regards,

Reiji Yoshimura MD, Ph.D

Professor and Chair

Department of Psychiatry, University of Occupational and Envuronmental Halth Japan, Kitakyushu, Fukuoka8078555, Japan 

Reviewer 3 Report

Authors have addressed my comments and concerns - nothing further required.

Author Response

To the reviewer#3

Thank you so much for kindly reviewing our manuscript.

Best regards,

Reiji Yoshimura MD, Ph.D

Professor and Chair

Department of Psychiatry, University of Occupational and Environmental Health Japan, Kitakyushu, Fukuoka8078555, Japan